# Right ventricle free wall longitudinal strain screening of lung transplant candidates

**Vittorio Scaravilli**[1,2], **Silvia Scansani**[1], **Paolo Meani**[3,4]*, **Gloria Turconi**[3], **Amedeo Guzzardella**[3], **Marco Bosone**[3], **Claudia Bonetti**[3], **Marco Vicenzi**[5,6], **Letizia Corinna Morlacchi**[7], **Valeria Rossetti**[7], **Lorenzo Rosso**[3,5], **Francesco Blasi**[3,7], **Mario Nosotti**[3,5], **Giacomo Grasselli**[1,2,3]

1 Department of Anesthesia, Critical Care and Emergency, Fondazione IRCCS Ca' Granda—Ospedale Maggiore Policlinico, Milan (MI), Italy, 2 Department of Biomedical, Surgical and Dental Sciences, University of Milan, Milan (MI), Italy, 3 Department of Pathophysiology and Transplantation, University of Milan, Milan (MI), Italy, 4 Faculty of Health, Medicine and Life Sciences, Maastricht University, Maastricht, The Netherlands, 5 Department of Cardio-thoraco-vascular diseases, Fondazione IRCCS Ca' Granda—Ospedale Maggiore Policlinico, Milan (MI), Italy, 6 Dipartimento di Scienze Cliniche e di Comunità, University of Milan, Milan (MI), Italy, 7 Department of Internal Medicine, Respiratory Unit and Cystic Fibrosis Center, Fondazione IRCCS Ca' Granda—Ospedale Maggiore Policlinico, Milan (MI), Italy

* paolo.meani@unimi.it

## Abstract

### Background

Lung transplant (LUTX) candidates have subclinical right ventricular (RV) dysfunction, which has not yet been assessed by speckle-tracking echocardiography (STE)-derived RV free-wall longitudinal strain (RVFWLS). To evaluate the prevalence of RV dysfunction by RVFWLS and its relationship with conventional RV echocardiographic indexes in LUTX candidates.

### Methods

In a single-center prospective observational cohort study, from January 2021 to March 2023 consecutive LUTX candidates underwent cardiac catheterization, radionuclide ventriculography, standard and STE. The diagnostic accuracy of RV ejection fraction by ventriculography (RVEF), tricuspid annular plane excursion (TAPSE), fractional area change (FAC), tricuspid peak annulus systolic velocity (S') versus RVFWS were computed.

### Results

Thirty-four patients (female, 41%) with a mean age of 48 [36–59] years old enlisted for pulmonary fibrosis (35%) and cystic fibrosis (30%) were included. At cardiac catheterization, only 7 (23%) had pulmonary hypertension. Around 15–25% presented right heart enlargement. Tricuspid regurgitation was present in 20 (60%) of the patients. Median RVFWLS was -20.1% [-22.5%–-17%], being impaired (> -20%) in 16 (47%) of the patients. RVFWLS identified the highest percentage (47%) of RV dysfunction, compared to TAPSE (32%), S' (27%), FAC (26%), and ventriculography (15%), which had very low sensitivity for detecting RV dysfunction compared to RVFWLS.

**Data Availability Statement:** All relevant data are within the manuscript and its Supporting Information files.

**Funding:** The Fondazione per la Ricerca sulla Fibrosi Cistica supported the study (# FFC 27/2019), project adopted by: Delegazione FFC di Napoli San Giuseppe Vesuviano and Delegazione FFC di Como Dongo). This study was (partially) funded by Italian Ministry of Health – Current Research IRCCS.

**Competing interests:** The Fondazione per la Ricerca sulla Fibrosi Cistica supported the study (# FFC 27/2019), and in particular the project was adopted by the Delegazione FFC di Napoli San Giuseppe Vesuviano and the Delegazione FFC di Como Dongo). The first author Dr. Vittorio Scaravilli received from the aforementioned grant support for publication and congress participation. Prof. Giacomo Grasselli received payment for lectures from Thermo-Fisher and Pfizer Pharmaceuticals and travel-accommodation-congress support from Biotest (all these relationships are unrelated with the present work). Moreover, this study was (partially) funded by Italian Ministry of Health – Current Research IRCCS. This does not alter our adherence to PLOS ONE policies on sharing data and materials.

## Conclusions

In patients enlisted for LUTX, RV dysfunction assessed by STE-derived RVFWLS is highly prevalent. STE can detect RV dysfunction better than standard two-dimensional echocardiography and ventriculography. Further studies are urgently needed to define the clinical implications and the prognostic value of RV dysfunction measured with RVFWLS.

## Introduction

Bilateral Lung transplantation (LUTX) is a viable option for selected patients facing end-stage respiratory failure [1]. This condition often comes with complications such as pulmonary hypertension, secondary right ventricular hypertrophy, and chronic right heart failure [2]. These complexities are frequently observed in patients listed for primary pulmonary hypertension [3]. Moreover, individuals with varying indications for LUTX commonly experience subtle right ventricular (RV) dysfunction [4, 5]. During the surgical procedure, factors like hypoxia, hypercapnia, and particularly pulmonary artery sequential cross-clamping abruptly impact RV contractility, exacerbating RV afterload. This unsettles the already unstable chronic equilibrium between increased afterload and RV hypertrophy, leading to acute RV failure [6]. In such cases, emergent extracorporeal life support (ECLS) becomes the sole available rescue therapy [7], albeit not without potential adverse effects. There include the activation of pro-inflammatory cascade due to blood-circuit contact [8], heightened demand for allogenic blood components, and an increased risk of primary graft dysfunction [9]. Accurate risk stratification for RV dysfunction could significantly enhance perioperative hemodynamic management and allow judicious use of ECLS for highly selected cases.

The established gold standards for pre-operative cardiac evaluation in LUTX candidates involve radionuclide ventriculography [10] or cardiac magnetic resonance [11], both of which, while highly accurate, are not bedside accessible and entail logistical complexities. Additionally, radionuclide ventriculography raises concerns due to radiation exposure. Conversely, echocardiography offers a comprehensive, non-invasive, and repeatable assessment of cardiac function and is extensively used for evaluating the pre-operative cardiac function in these patients.

However, conventional echocardiography [12], relying on geometric assumptions (as the fractional area change—FAC) or extrapolation of the shortening from one single point of the ventricle (as the tricuspid annular plane excursion—TAPSE), exhibits limited diagnostic capability in assessing RV function due to the unique crescent-like shape of the right ventricle. Conversely, speckle-tracking echocardiography, specifically the RV free-wall longitudinal strain (RVFWLS) has demonstrated high diagnostic accuracy in detecting RV dysfunction [13]. Despite this, RVFWLS has rarely been assessed and compared to other indexes of RV function in patients awaiting LUTX [14].

We hypothesized that patients enlisted for LUTX may have impaired RVFWLS, and that RVFWLS can have better diagnostic capabilities in detecting RV dysfunction as compareed to standard two-dimensional echocardiography and ventriculography. Accordingly, this prospective observational cohort study aimed to: 1) evaluate the RVFWLS in patients listed for LUTX; 2) investigate the relationship between RVFWLS and conventional RV echocardiographic indexes.

## Methods

The study received approval from the Institutional Ethics Committee (Comitato Etico Milano Area 2, # 754_2019) and was registered at clinicaltrials.gov under the identifier NCT05855148. Written informed consent was obtained. Furthermore, adherence to the most recent International Society of Lung Transplant Society ethics statement [15] and compliance with STROBE guidelines [16] were ensured while reporting this study.

This study is a single-center prospective observational cohort analysis of consecutive patients enlisted for LUTX at an Italian tertiary referral center from January 2021 to November 2022. Due to logistical constraints subsequent to the COVID-19 pandemic, actual recruitment of patients commenced on the 1st November 2021, and ended on the 31st November 2022.

All patients enlisted for LUTX during the study period were considered for inclusion. Exclusion criteria were: 1) single LUTX; 2) re-transplantation; 3) patients bridged to LUTX with veno-venous ECLS; 4) incomplete medical records. At our Institution, patients undergo a comprehensive cardiac evaluation at LUTX enlistment, comprising: 1) invasive right heart catheterization; 2) multi-gated radionuclide ventriculography; 3) transthoracic echocardiography performed by a specialized cardiologist. For in-depth details on LUTX management of at our Institution, refer to the Additional Methods in S1 File.

To conduct this study, following the enlistment approval, the research team contacted the patients, and a specialized sonographer (SS) and a specialized cardiologist (PM), blinded echocardiography at enlistment, performed an additional echocardiographic examination to measure RV strain by STE. Utilizing a GE Vivid E95 ultrasound machine (GE Healthcare, Milwaukee, WI), images were acquired during breath holds with stable electrocardiographic recordings and stored digitally for subsequent offline analysis using EchoPAC Clinical Workstation Software (GE Healthcare, Milwaukee, WI). RV global longitudinal strain (RVGLS) and RV free wall longitudinal strain (RVFWLS) were calculated using the two-dimensional RV-focused view [17, 18] or subcostal views [19] when the formed was inaccessible due to anatomic malformations (see S1 File for additional details). Based on the most recent available data [20], patients were categorized as having impaired ($>$-20%) or normal ($<$ -20%) RVFWLS.

Additionally, measurements according to the international guidelines [12, 21] were obtained, on the same frame were STE-derived measurements were obtained: right atrium area, RV end-diastolic area, FAC, TAPSE, tissue Doppler imaging tricuspid peak annulus systolic velocity (S'), and pulmonary artery systolic pressure.

The following data were prospectively collected at the time of LUTX enlistment: demographics, weight, height, LUTX indication (further aggregated in pulmonary fibrosis vs. not pulmonary fibrosis), comorbidities, lung allocation score (LAS), oxygen requirement at rest, spirometry, arterial blood gas analyses, diffusing capacity of carbon monoxide, six-minute walking test, pulmonary arterial pressures, incidence of pulmonary hypertension (i.e., mean pulmonary artery pressure $>$ 25 mmHg) and cardiac output (by invasive cardiac catheterization); pulmonary scintigraphy; right ventricle ejection fraction (RVEF) measured by multi-gated radionuclide ventriculography.

### Data analysis

Sample size calculation was performed utilizing a Wilcoxon signed-rank test (one sample case). Considering a normal value of RVFWLS of -29% ± 4.5% [12], with an α error probability of 0.05, and a power (1-β error probability) of 0.8, a reduction in RVFWLS in patients enlisted for LUTX consisting with a RVFWLS of -27 could be detected with a sample size of 35.

Data were reported as the median [first-third quartile] and number of events (percentage of the subgroup) for continuous and categorical variables, respectively. Missing data were not imputed, and, whereas strain measurement where missing (i.e., patients with poor acoustic windows for RV evaluation), were not considered for the echocardiographic analysis. The Z-test was utilized to compare the patients' population with normality values [12, 21, 22]. The correlation between continuous variables was tested with the $R^2$ linear regression. Sensitivity, specificity, positive predictive value (PPV), negative predictive values (NPV), and associated confidence intervals (CI) [23] of TAPSE, FAC, S', and RVEF by multi-gated radionuclide ventriculography vs. RVFWLS were computed. Comparison between patients' cohorts (i.e., normal RVFWLS vs. compromised RVFWLS) was performed with $Chi^2$ or Fisher Exact Test. Logistic regression was utilized to assess the possible association between enlistment patients' characteristics and patients' cohort. The odds ratios (OR) and associated 95% likelihood ratio-based confidence intervals were calculated. Statistical significance was accepted at $P < 0.05$. The JMP® pro 16.0 (SAS, Cary, NC) was utilized.

## Results

Between January 2021 to March 2023, 64 consecutive patients were enlisted for lung transplantation at our Institution. Among them, 44 met the study's inclusion criteria (see Fig 1). Among these, 10 (23%) patients had poor acoustic windows for RV echocardiographic evaluation (see S1 Table in S1 File), resulting in 34 patients included in the analysis (see Table 1). No missing data was documented in the cohort of included patients. Patients with poor acoustic windows showed a higher asymmetric pulmonary scintigraphy (p = 0.002, OR 0.87 (0.76–0.99) compared to those with good acoustic windows. Challenging acoustic windows were primarily found in patients with thoracic and mediastinal anatomical alterations.

Enrolled patients were predominantly male (59%), with a median age of 48.0 [36.0–59.0], with a relatively high LAS (i.e., 38.2 [34.9–42.6]). The majority enlisted for lung transplantation due to pulmonary fibrosis (35%). The second most common indication to LUTX was cystic fibrosis (30%), while none were enlisted for primary pulmonary hypertension. Cardiac

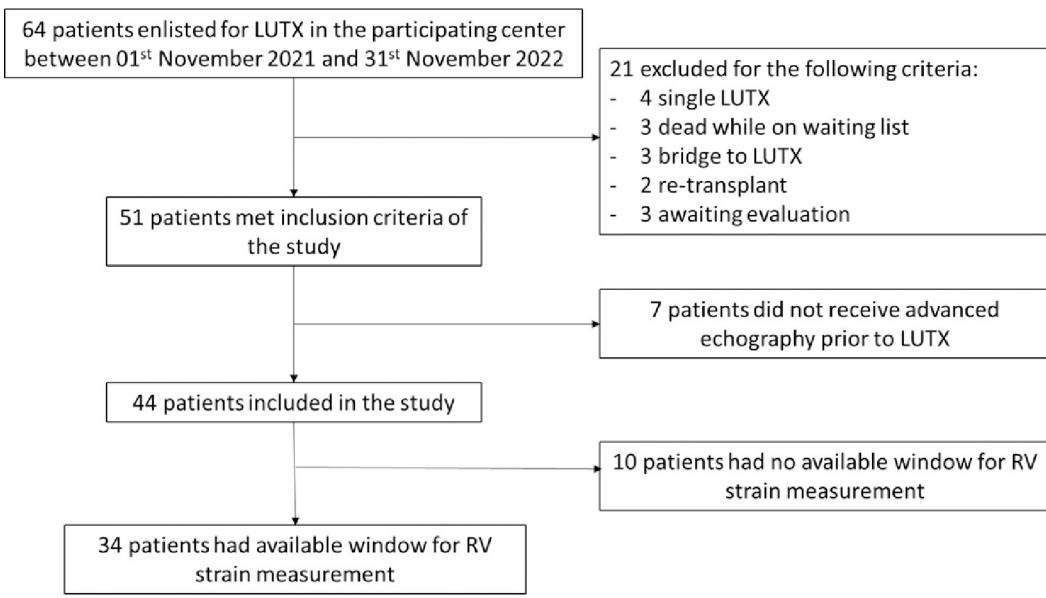

**Fig 1. Patients recruitment flowchart.** LUTX: lung transplantation. RV: right ventricle.

**Table 1. Patients' characteristics.**

| | | |
|---|---|---|
| Gender (Female) | | 14 (41%) |
| Age at enlistment (years) | | 48.0 [36.0–59.0] |
| Weight (kg) | | 62.5 [44.8–72.3] |
| Height (cm) | | 169 [158–175] |
| BMI (kg/m$^2$) | | 22.5 [18.2–25.8] |
| BSA (m$^2$) | | 1.7 [1.4–1.8] |
| Diagnosis | Pulmonary Fibrosis | 12 (35%) |
| | Cystic Fibrosis/bronchiectasis | 10 (29%) |
| | Chronic Obstructive Pulmonary Disease | 9 (26%) |
| | Other | 2 (6%) |
| | Hypersensitivity pneumonitis | 1 (3%) |
| Lung Allocation Score | | 38.2 [34.9–42.6] |
| O$_2$ need at rest | | 1.0 [0.8–2.0] |
| | FVC (% predicted) | 49.0 [38.8–68.0] |
| | FEV$_1$ (% predicted) | 28.5 [18.8–40.3] |
| | DLCO (% predicted) | 23.0 [13.8–36.0] |
| 6 Minutes Walking Test (mt) | | 383 [252–457] |
| Arterial Blood Gas Analyses at rest | FiO$_2$ at rest | 24 [21–28] |
| | pH | 7.44 [7.41–7.47] |
| | pO$_2$ (mmHg) | 76.5 [69.0–88.5] |
| | pCO$_2$ (mmHg) | 44.0 [37.8–51.3] |
| | HbO$_2$ (%) | 95.2 [94.0–98.0] |
| Cardiac Catheterization | CO (L/min) | 4.8 [4.3–6.2] |
| | CI (L/min/m$^2$) | 3.0 [2.6–3.3] |
| | HR (bpm) | 73.5 [65.5–87.8] |
| | PAPm (mmHg) | 21.0 [18.0–24.0] |
| | Pw (mmHg) | 9.0 [6.0–10.3] |
| Pulmonary hypertension | | 7 (23%) |
| Pulmonary Scintigraphy (% left lung) | | 47.0 [36.4–51.0] |
| RV Ejection Fraction (%)* | | 49 [45–62] |
| RV Ejection Fraction < 40%* | | 5 (15%) |

Data are presented as absolute frequency (% of the included patients) or as median and interquartile range. BMI, body mass index; BSA, body surface area; FEV$_1$, 1st second forced expiratory volume; FVC, forced vital capacity; DLCO, diffusing capacity of the lungs for carbon monoxide; pO$_2$, Oxygen Partial Pressure; pCO$_2$, carbon dioxide partial pressure; HbO$_2$, hemoglobin saturation; CO, cardiac output; CI, cardiac index; HR, heart rate; PAPm, mean pulmonary artery pressure; Pw, wedge pressure; RV, right ventricle

*) derived by multi-gated radionuclide ventriculography.

catheterization indexes at enlistment fell within the normal range: only 7 (23%) patients had secondary pulmonary hypertension, and 5 (15%) had an impaired RV ejection fraction at multi-gated radionuclide ventriculography.

RV chamber sizes are reported in Table 2. Compared to the gender-matched and BSA-matched normal population, these patients showed statistically significant smaller atria, with larger RV diameters and smaller longitudinal dimensions, leading to larger RV end-systolic and end-diastolic areas. Around 15–25% showed abnormally enlarged right heart chambers. Tricuspid regurgitation was absent in 13 patients (39%), mild in 17 patients (51%), and moderate in 3 patients (9%). Pulmonary artery systolic pressure was obtained in only 13 (38%) patients.

**Table 2. Morphologic right heart echocardiographic data.**

| Index | Normal range | | Measured | | P value Z-test | |
|---|---|---|---|---|---|---|
| | **male** | **female** | **male** | **female** | | |
| Right atrium area (cm²) | 14 ± 2 | | 12.3 [10.1–14.0] | | 0.031 | |
| Right atrium area/BSA(cm²/m²) | 8.3 ± 0.1 | | 7.2 [5.7–9.2] | | 0.082 | |
| Abnormal Right atrium area (cm²) | >18 | | 4 (12.5%) | | | |
| RV end-diastolic area (cm²) | 17 ± 3.5 | 14 ± 3 | 18.9 [17.5–24.4] | 13.9 [12.6–17.4] | < 0.001 | 0.096 |
| Abnormal RV end-diastolic area (n) | > 24 | > 20 | 6 (17.6%) | | | |
| RV end-diastolic area/BSA (cm/m²) | 8.8 ± 1.9 | 8.0 ± 1.7 | 10.5 [8.9–13.2] | 9.6 [7.9–11.4] | < 0.001 | < 0.001 |
| Abnormal RV end-diastolic area/BSA (n) | > 12.6 | > 11.5 | 9 (26.4%) | | | |
| RV end-systolic area (cm²) | 9 ± 3 | 7 ± 2 | 12.0 [9.1–16.3] | 8.0 [6.2–9.2] | < 0.001 | < 0.001 |
| Abnormal RV end-systolic area (n) | > 15 | > 11 | 8 (23.5%) | | | |
| RV end-systolic area/BSA (cm²/m²) | 4.7 ± 1.35 | 4.0 ± 1.2 | 6.5 [5.1–8.6] | 5.1 [4.8–6.3] | < 0.001 | < 0.001 |
| Abnormal RV end-systolic area/BSA (n) | > 7.4 | > 6.4 | 10 (29.4%) | | | |
| RV basal diameter (mm) | 33 ± 4 | | 35.0 [31.8–40.3] | | < 0.001 | |
| Abnormal RV basal diameter (n) | > 41 | | 7 (20.6%) | | | |
| RV mid-cavitary diameter (mm) | 27 ± 4 | | 28.5 [25.0–35.5] | | < 0.001 | |
| Abnormal RV mid-cavitary diameter (n) | > 35 | | 8 (23.5%) | | | |
| RV longitudinal dimension (mm) | 71 ± 6 | | 62.0 [56.3–69.5] | | < 0.001 | |
| Abnormal RV longitudinal diameter (n) | > 84 | | 0 (0.0%) | | | |

RV, right ventricle; BSA; body surface area. Of note, for RV areas, different normal ranges are described for male and female patients.

Table 3 shows the conventional echocardiographic indexes used for RV function evaluation. Compared to the gender-matched and BSA-matched normal population, our patients' cohort had a statistically significant impaired systolic function. Approximately 30% presented an impaired RV contractility according to the conventional echocardiographic values.

Median RV global longitudinal strain (RVGLS) was -17.6% [-19.5%–-14.4%], and RV free wall longitudinal strain (RVFWLS) was -20.1% [-22.5%–-17%]. In comparison to the normal population, RVGLS and RVFWLS were significantly impaired in the study cohort (p<0.001), Only 18 (52%) patients had RVFWLS within the normal reference interval (i.e., < -20%), while it was impaired in the remaining 16 (47%). Fig 2 depicts a case of a patient enlisted to LUTX, documenting an almost normal conventional echocardiographic values associated with altered RVFWLS.

The linear correlation between RVGLS and RVFWLS had p<0.001 and $R^2$ = 0.934) (see S1 Fig in S1 File). However, despite statistically significant, the correlation between RV strain and

**Table 3. Conventional right ventricle systolic function echocardiographic data.**

| Index | Range | Measured | P value Z-test |
|---|---|---|---|
| TAPSE (mm) | 23 ± 3.5 | 20.0 [16.0–22.0] | < 0.001 |
| Abnormal TAPSE (mm) | < 17 | 11 (32.4%) | |
| S' velocity RV (cm/s) | 15 ± 2.5 | 11.0 [9.0–13.0] | < 0.001 |
| Abnormal S' velocity RV (cm/s) | < 9.5 | 9 (27.3%) | |
| Fractional Area Change (%) | 49 ± 7 | 41.0 [28.5–47.3] | < 0.001 |
| Abnormal Fractional Area Change (%) | < 35 | 9 (26.5%) | |

RV, right ventricle; TAPSE; tricuspid annular plane systolic excursion; S', tissue Doppler positive peak systolic wave velocity.

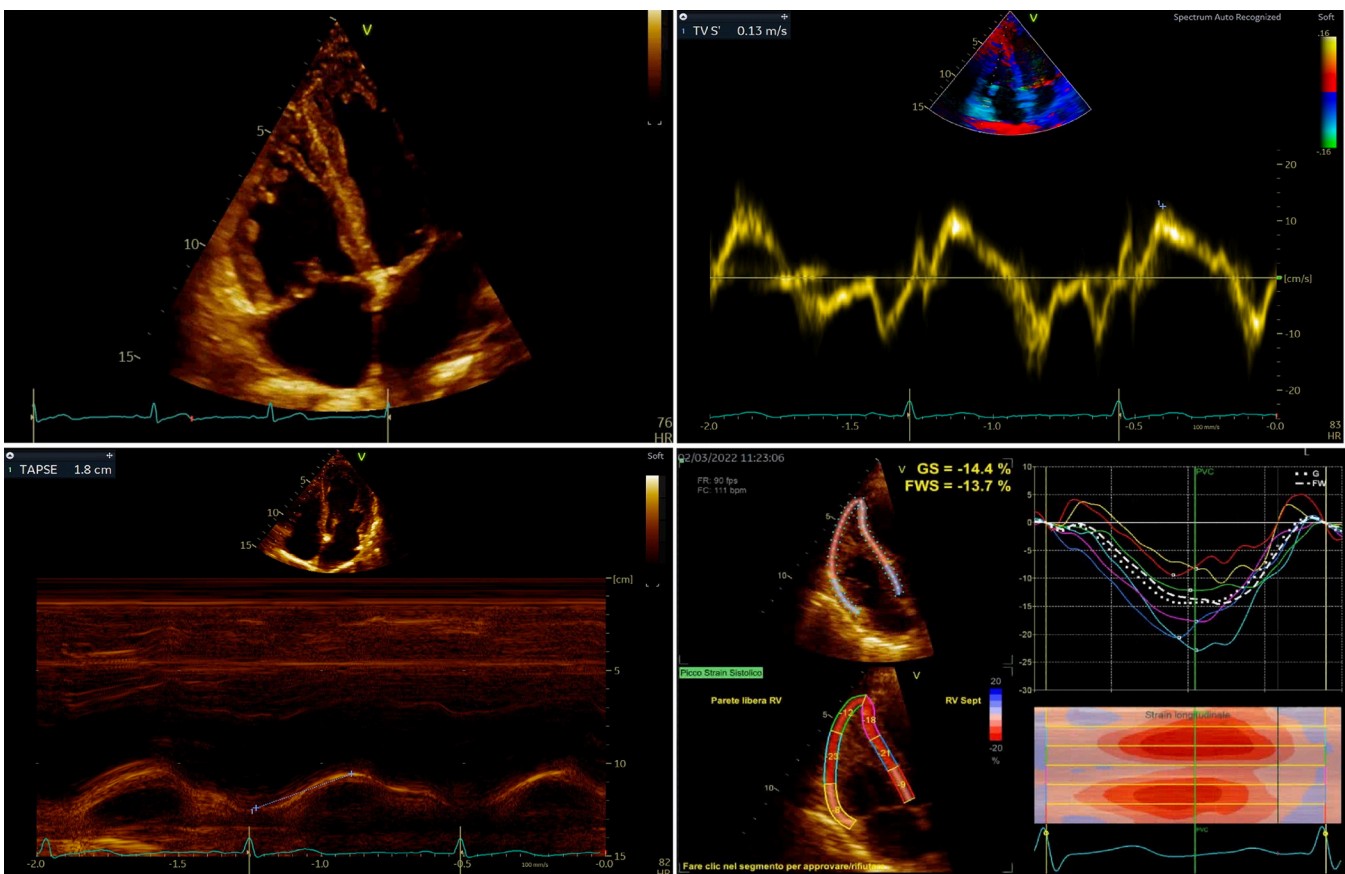

**Fig 2. The superior diagnostic capabilities of strain imaging in detecting right ventricular systolic dysfunction compared to standard two-dimensional echocardiography in a lung transplant candidate.** The apical RV-focused view is depicted in the left upper panel, which is central for both two-dimensional and speckle-tracking echocardiographic imaging evaluation. On the right upper panel, the right ventricle tissue Doppler imaging is shown, and the peak systolic velocity wave (S') is measured. As in most of the lung transplant candidates in our cohort, S' value is normal (13 cm/s). On the lower left panel, M-mode imagining on the tricuspid annular plane excursion (TAPSE) is shown. Similarly to S', the TAPSE value is normal (18 mm). On the right lower panel, speckle-tracking echocardiography of the right ventricle was carried out, and the RV free wall longitudinal strain (RWFLS) was calculated with a significantly impaired value (-13.7%).

standard RV echocardiographic indexes exhibited weak associations (see Fig 3). All strain measurements showed poor correlation coefficient with FAC and S' ($R^2$ approximately 0.1), while a slightly better $R^2$ was observed between RV strain and TAPSE (i.e., 0.3). No statistically significant correlation was identified between RV ejection fraction (by radionuclide ventriculography) and RV strain, specifically RVGLS and RVFWLS (RVGLS: $R^2 = 0.058$, p = 0.199; RVFWLS: $R^2 = 0.023$, p = 0.403) (see S2 Fig and Additional Results in S1 File). The linear correlation between RVFWLS, FAC, S', and TAPSE and hemodynamic parameters obtained from right heart catheterization is depicted in S3-S6 Figs in S1 File. No statistically significant correlation was observed between RVFWLS and cardiac output ($R^2 = 0.116$, p = 0.061). Contrarily, we observed a statistically significant correlation between RVFWLS and cardiac index ($R^2 = 0.245$, p = 0.005), PAPm ($R^2 = 0.512$, p<0.001), pulmonary artery wedge pressure ($R^2 = 0.119$, p = 0.013), and pulmonary resistances ($R^2 = 0.398$, p<0.001). Compared to FAC, S', and TAPSE, RVFWLS showed the strongest linear correlations.

Fig 4 presents the distribution of RV function impairment based on multi-gated radionuclide ventriculography, conventional (FAC, S' and TAPSE), and advanced echocardiographic indexes (RVFWLS). Notably, RV-free wall longitudinal strain identified the highest percentage

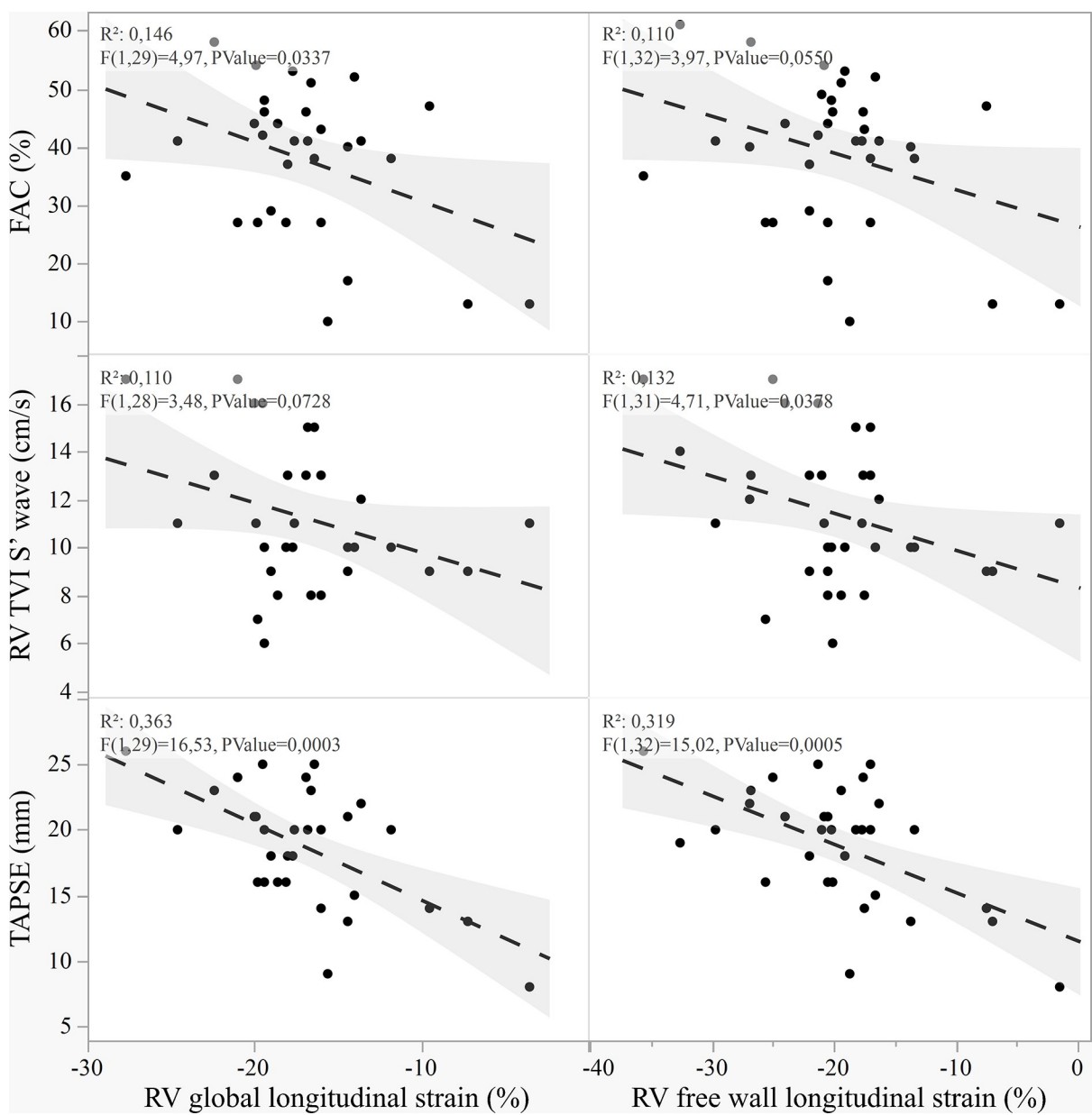

**Fig 3. Linear correlations between right ventricle systolic strain and standard RV systolic echocardiographic analyses.** FAC, fractional area change; RV TVI S', pulsed-wave tissue Doppler imaging tricuspid peak annulus systolic velocity; TAPSE, tricuspid annular plane excursion; RV, right ventricle.

of RV impairment (47%) compared to TAPSE (32%), S' (27%), FAC (26%), and ventriculography (15%).

RVEF obtained through ventriculography and standard RV echocardiographic analysis (i.e., TAPSE, S', and FAC) demonstrated very low sensitivity in detecting RV dysfunction compared to RVFWLS, see Table 4. Overall, standard tests showed a poor performance in identifying right ventricular dysfunction.

The main risk factors for RV dysfunction at enlistment are reported in Table 5. Males and higher BMI patients had statistically significantly higher odds for impaired RVFWLS. Moreover, patients with lower cardiac index and higher pulmonary arterial pressures had

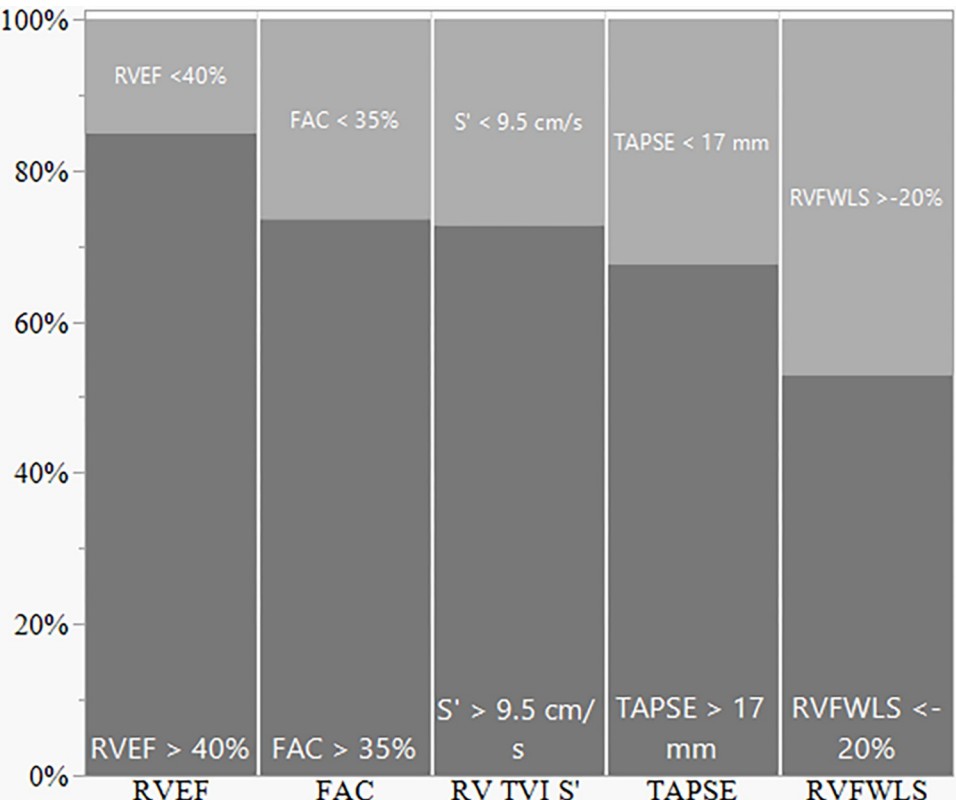

**Fig 4. Distribution of right ventricular systolic impairment according to different measures.** RVEF, right ventricle ejection fraction derived by multi-gated radionuclide ventriculography; FAC, fractional area change; RV TVI S', pulsed-wave tissue Doppler imaging tricuspid peak annulus systolic velocity; TAPSE, tricuspid annular plane excursion, RVFWLS, right ventricle free wall longitudinal strain.

statistically significantly increased RV dysfunction risk. Multivariate models were not considered appropriate, given the limited number of events.

## Discussion

In this prospective observational study, we described the ability of speckle-tracking echocardiography in detecting pre-operative RV dysfunction and its association with different conventional techniques in a cohort of adults enlisted for double lung transplantation (LUTX). RVFWLS was altered in almost half of the patients in our cohort, while conventional techniques showed RV dysfunction in fewer patients.

**Table 4. Performance of conventional right ventricle systolic function echocardiographic tests vs. right ventricle free wall longitudinal strain.**

| Index | Pathologic Range | p | Sensitivity (95% CI) | NPV (95% CI) | Specificity (95% CI) | PPV (95% CI) |
|---|---|---|---|---|---|---|
| RVEF | <40 (%) | 0.478 | 0.11 (0.03–0.32) | 0.42 (0.26–0.6) | 0.80 (0.54–0.93) | 0.40 (0.11–0.76) |
| TAPSE | < 17 (mm) | 0.179 | 0.22 (0.09–0.45) | 0.39 (0.22–0.59) | 0.56 (0.33–0.78) | 0.36 (0.15–0.64) |
| S' | < 9.5 (cm/s) | 0.943 | 0.28 (0.12–0.51) | 0.45 (0.28–0.65) | 0.73 (0.48–0.89) | 0.55 (0.26–0.81) |
| FAC | < 35 (%) | 0.854 | 0.27 (0.12–0.51) | 0.48 (0.30–0.66) | 0.75 (0.50–0.89) | 0.55 (0.26–0.81) |

NPV, negative predictive value; PPV, positive predictive value; RVEF, right ventricle ejection fraction derived by multi-gated radionuclide ventriculography; TAPSE; tricuspid annular plane systolic excursion; S', tissue Doppler positive peak systolic wave velocity; FAC, fractional area change.

**Table 5. Risk factors for impaired right ventricle function at enlistment.**

| | | RVFWLS <-20% (n = 18, 53%) | RVFWLS >-20% (n = 16, 47%) | p | OR (95% CI) |
|---|---|---|---|---|---|
| Gender (Male) | | 7 (39%) | 13 (81%) | **0.010** | **6.80 (1.41–32.8)** |
| Age at enlistment (years) | | 47.5 [33.7–59.0] | 49.5 [36.2–59.8] | 0.562 | 1.01 (0.96–1.06) |
| BMI (kg/m$^2$) | | 18.6 [17.7–23.5] | 24.2 [22.3–27.2] | **0.007** | **1.28 (1.05–1.56)** |
| Diagnosis | Pulmonary Fibrosis | 4 (22%) | 8 (50%) | 0.356 | — |
| | Cystic Fibrosis/bronchiectasis | 7 (39%) | 3 (18%) | | |
| | Chronic Obstructive Pulmonary Disease | 5 (27%) | 4 (25%) | | |
| | Other | 2 (10%) | 1 (6%) | | |
| Lung Allocation Score | | 37.8 [34.5–40.7] | 39.4 [36.1–43.4] | 0.681 | 1.02 (0.92–1.12) |
| O$_2$ need at rest (L/min) | | 1 [1–2] | 1 [0–2] | 0.424 | 1.21 (0.74–1.97 |
| Pulmonary Function Test | FVC (% predicted) | 49.0 [37.8–66.2] | 47.5 [38.7–73.7] | 0.875 | 0.99 (0.96–1.02) |
| | FEV$_1$ (% predicted) | 28.5 [18.5–39.3] | 28.0 [18.5–40.7] | 0.609 | 1.00 (0.97–1.03) |
| | DLCO (% predicted) | 22.5 [13.3–32.5] | 23.0 [13.7–47.7] | 0.649 | 1.00 (0.96–1.05) |
| 6 Minutes Walking Test (mt) | | 398 [255–451] | 380 [250–460] | 0.986 | 0.99 (0.99–1.00) |
| Blood Gas Analysis at rest | pH | 7.44 [7.41–7.46] | 7.44 [7.42–7.47] | 0.967 | — |
| | pO$_2$ (mmHg) | 76.5 [70.0–96.0] | 75.5 [66.7–80.7] | 0.159 | 0.96 (0.91–1.01) |
| | pCO$_2$ (mmHg) | 45.0 [41.0–51.3] | 39.0 [37.5–51.7] | 0.244 | 0.96 (0.89–1.03) |
| Cardiac Catheterization | Cardiac Output (L/min) | 4.8 [4.3–6.4] | 4.9 [4.0–5.6] | 0.213 | 0.68 (0.36–1.27) |
| | Cardiac Index (L/min/m$^2$) | 3.2 [2.7–3.7] | 2.7 [2.2–3.1] | **0.004** | **0.13 (0.02–0.75)** |
| | PAPm (mmHg) | 20 [17–23] | 22 [19–30] | **0.025** | **1.12 (0.98–1.28)** |
| | Pw (mmHg) | 6 [5–9] | 10 [6–11] | 0.156 | 1.19 (0.92–1.54) |
| Pulmonary hypertension (PAPm > 25 mmHg) | | 2 (11%) | 5 (35%) | 0.110 | 4.16 (0.66–26.1) |
| Pulmonary Scintigraphy (% left lung) | | 49 [33–53] | 42 [38–49] | 0.237 | 0.96 (0.90–1.02) |
| RV Ejection Fraction (%) * | | 52 [46–55] | 47 [43–50] | 0.077 | 0.92 (0.84–1.01) |
| RV Ejection Fraction < 40%* | | 2 (11%) | 3 (20%) | 0.478 | 2.00 (0.28–13.9) |

Data are presented as absolute frequency (% of the included patients) or as median and interquartile range. OR, odds ratio (per unit change in regressor for continuous variables). RVFWLS, right ventricle free wall longitudinal strain; BMI, body mass index; FVC, forced vital capacity; FEV$_1$, 1st second forced expiratory volume; DLCO, diffusing capacity of the lungs for carbon monoxide; pO$_2$, Oxygen Partial Pressure; pCO$_2$, carbon dioxide partial pressure; HbO$_2$, hemoglobin saturation; PAPm, mean pulmonary artery pressure; Pw, wedge pressure; RV, right ventricle

*) derived by multi-gated radionuclide ventriculography.

Assessing RV function in patients enlisted for LUTX is crucial [2], since the majority of them suffer from RV clinical or subclinical dysfunction [24] secondary to their respiratory failure. This dysfunction worsens during LUTX surgery due to factors such as pulmonary artery cross-clamping, hypoxia, and hypercapnia, causing a sudden and prolonged increase in RV afterload, negatively impacting RV performance. Since the management of intraoperative RV failure by extracorporeal circulation is prone to immediate and long-term consequences, stratifying RV function before LUTX is crucial.

The gold standards for RV evaluation (i.e., ventriculography and magnetic resonance) are costly, invasive, and have high logistical requirements and, thus, are generally performed only at LUTX enlistment. This significantly limits their usefulness during the perioperative period, since the majority of patients enlisted for LUTX have a progressive lung function worsening, possibly leading to progressive RV dysfunction. In other words, in this cohort, the gold standards provide a very high-quality evaluation of RV dysfunction, but cannot capture the evolution of the RV impairment while enlisted, during and after the surgical procedure.

Among the available techniques to detect RV dysfunction, echocardiography stands out as a non-invasive, safer, cost-effective, and more readily available method compared to other

cardiac assessment techniques [24–27]. However, conventional two-dimensional echocardiography indexes for RV assessment have limited accuracy due to the RV's unique morphology, and could be technically challenging [28]. In our study, standard echocardiography demonstrated by larger sub-pathological RV diameters and smaller longitudinal dimensions, with overall increased RV end-systolic and end-diastolic areas. Nearly one in four patients exhibited an abnormally enlarged RV, impacting tricuspid valve continence and leading to detected valvular regurgitation in around 60% of our cohort. In this setting, conventional echocardiographic parameters such as TAPSE, FAC, and S' have limited accuracy [29].

Speckle tracking echocardiography (STE) overcomes these limitations by assessing RV function through evaluating myocardial fibers' active shortening, less dependent on geometry, insonation angle, or operator inter-variability [30]. RV global longitudinal strain (RVGLS) and RV free wall longitudinal strain (RVFWLS) are prominent STE-derived RV strain parameters, demonstrating predictive roles in various cardiovascular conditions (i.e., pulmonary hypertension, heart failure, and valvular diseases) [31], but have never been assessed in LUTX candidates.

A particular limitation of speckle-tracking echocardiography is that it requires optimal and high frame rate captures [32], which might be challenging in patients with poor acoustic windows, such as patients with thoracic and mediastinal anatomical alterations, as LUTX candidates [33]. Despite these limitations, we were able to perform STE RV assessment in up to 77% of patients enlisted for LUTX. Of note, specialized and expert operators performed all the echocardiographic assessments in an echocardiography laboratory, thus possibly limiting this approach to everyday practice. However, technological advancements and newer devices may improve feasibility and overcome existing echocardiographic limitations [34].

RV dysfunction was found in 15%, 30%, and 47% of the patients' cohort by ventriculography, conventional echocardiography, and speckle-tracking echocardiography (RVFWLS), respectively. Notably, RVFWLS detected RV dysfunction in a significant proportion of patients undetected by standard RV echocardiography. Therefore, in our study, TAPSE, FAC, S' as well as ventriculography exhibited poor sensitivity in detecting RV dysfunction compared to RVFWLS. This is reasonably because speckle-tracking echocardiography does not rely on geometric assumptions (as FAC), nor extrapolates the RV shortening from one single point of the ventricle (as TAPSE) [35].

Among the STE-derived variables obtained, RVFWLS would better unmask RV dysfunction in our population than RVGLS. RVGLS indicates the average strain value of both the RV free wall and septal segments, whereas RVFWLS specifically represents the average strain value of the RV free wall segments exclusively [36]. Given that both ventricles share the interventricular septum, the left ventricle contributes 20% to 40% of the primary force for RV contraction. Consequently, RVFWLS stands as a more representative indicator of pure RV dysfunction.

In addition to the technical advantages provided by STE, RV strain assessment might further improve the standard LUTX pre-operative work-up. This bedside, highly accurate technique could be periodically performed, especially during acute lung deterioration. However, at present, STE-derived RV strain should not take over radionuclide ventriculography or cardiac magnetic resonance, which remain the gold standards for pre-operative cardiac evaluation in LUTX candidates. Of note, compared to FAC, S', and TAPSE, RVFWLS showed stronger linear correlations with hemodynamic parameters obtained during cardiac catheterization. This further strengthens the argument in favor of utilizing STE-derived measurements for repeated, periodical, non-invasive assessment of RV function of patients awaiting LUTX.

Notably, the 2021 international consensus recommendations for anesthesiologic and intensive care management in lung transplantation did not set recommendations for RV failure risk stratification [37]. However, the increased availability and ongoing advancements in

techniques to detect RV dysfunction, like STE-derived strain analysis, may help bridge this gap in knowledge. Our findings stress the need for further studies examining clinical outcomes in LUTX candidates identified with RV dysfunction via RV strain. To date, literature on this topic remains limited. In a seminal paper, Kusunose et al. [14] analyzed the prognostic value of STE-derived RV strain in patients undergoing LUTX but found no association between pre-operative RV strain and mortality. Contrarily to our work, the study was retrospective and possibly biased by including a selected cohort of 89 patients (among an overall cohort of 933 patients) with a cardiac history prior to LUTX. The timespan covered by the study ranges from 2001 to 2012, and thus, it cannot describe the current clinical course of LUTX patients. Moreover, the follow-up time to assess the impact on outcomes ranged between 5 to 143 months. Such an assessment is unfeasible in our cohort since most of the patients included in the study at the time of this writing were yet to be transplanted. We will actively pursue the chance of assessing the impact on short and long-term outcomes of the patients included in the present study.

Despite these novel findings, several limitations exist. We acknowledge the limitations of statistical power in our analysis, leading to ample confidence intervals reflecting the exploratory nature and small sample size of our study. Further, more populated studies are necessary to confirm our findings. Again, the study's relatively small, single-center cohort, and reliance on highly trained personnel in a specialized center might limit immediate applicability to other centers. Additionally, the absence of routine cardiac magnetic resonance in our center prevented validation of STE-derived RV strain against the current gold standard. Measurement of STE-derived strain analysis was not possible due to missing optimal echocardiographic windows in almost 25% of the patients. Further technological developments are necessary to improve its routine clinical application in patients enlisted for LUTX. Finally, clarifying the prognostic implications of pre-operative RV assessment remains crucial.

## Conclusions

This prospective observational stud describes STE- derived RVFWLS in LUTX candidates and its association with standard echocardiographic measurements and multi-gated radionuclide ventriculography. The findings suggest STE's feasibility and enhanced capability in detecting RV dysfunction compared to standard techniques. Further research is urgently needed to define clinical implications, particularly in intraoperative and postoperative outcomes.

## Supporting information

**S1 Checklist.**
(DOC)

**S1 File. Additional methods, and results.**
(DOCX)

**S2 File. Study protocol.**
(DOC)

**S1 Dataset. Complete anonymized dataset.**
(RAR)

## Author Contributions

**Conceptualization:** Vittorio Scaravilli.

**Data curation:** Vittorio Scaravilli, Silvia Scansani, Paolo Meani, Gloria Turconi, Amedeo Guzzardella, Marco Bosone, Claudia Bonetti, Marco Vicenzi, Letizia Corinna Morlacchi, Valeria Rossetti.

**Formal analysis:** Vittorio Scaravilli, Silvia Scansani, Paolo Meani, Amedeo Guzzardella, Marco Vicenzi.

**Funding acquisition:** Vittorio Scaravilli.

**Investigation:** Vittorio Scaravilli, Paolo Meani, Gloria Turconi, Amedeo Guzzardella, Marco Bosone, Claudia Bonetti, Marco Vicenzi, Letizia Corinna Morlacchi.

**Methodology:** Vittorio Scaravilli, Silvia Scansani, Paolo Meani, Gloria Turconi, Marco Bosone, Claudia Bonetti, Marco Vicenzi.

**Project administration:** Vittorio Scaravilli, Gloria Turconi, Amedeo Guzzardella, Valeria Rossetti.

**Resources:** Vittorio Scaravilli, Silvia Scansani, Marco Vicenzi, Valeria Rossetti, Lorenzo Rosso, Mario Nosotti, Giacomo Grasselli.

**Software:** Silvia Scansani.

**Supervision:** Vittorio Scaravilli, Gloria Turconi, Lorenzo Rosso, Francesco Blasi, Mario Nosotti, Giacomo Grasselli.

**Validation:** Vittorio Scaravilli, Francesco Blasi, Mario Nosotti, Giacomo Grasselli.

**Visualization:** Vittorio Scaravilli, Lorenzo Rosso.

**Writing – original draft:** Vittorio Scaravilli, Silvia Scansani, Paolo Meani, Letizia Corinna Morlacchi, Lorenzo Rosso, Francesco Blasi, Mario Nosotti, Giacomo Grasselli.

**Writing – review & editing:** Vittorio Scaravilli, Paolo Meani, Lorenzo Rosso, Giacomo Grasselli.

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
