## [Decision Letter · Decision Letter 0]

20 Sep 2024

PONE-D-23-43919Right Ventricle Free Wall Longitudinal Strain Screening of Lung Transplant CandidatesPLOS ONE

Dear Dr. Meani,

Thank you for submitting your manuscript to PLOS ONE. After careful consideration, we feel that it has merit but does not fully meet PLOS ONE’s publication criteria as it currently stands. Therefore, we invite you to submit a revised version of the manuscript that addresses the points raised during the review process.

We look forward to receiving your revised manuscript.

Kind regards,

Yashendra Sethi

Academic Editor

PLOS ONE

3. Thank you for stating the following in the Competing Interests section: [VS received support for publication and congress participation by Fondazione Ricerca Fibrosi Cistica supported the study (# FFC 27/2019).GG received payment for lectures from Thermo-Fisher and Pfizer Pharmaceuticals and travel-accommodation-congress support from Biotest (all these relationships are unrelated with the present work). The remaining authors do not have any potential conflicts of interest.  ]. Please confirm that this does not alter your adherence to all PLOS ONE policies on sharing data and materials, by including the following statement: "This does not alter our adherence to PLOS ONE policies on sharing data and materials.” (as detailed online in our guide for authors http://journals.plos.org/plosone/s/competing-interests). If there are restrictions on sharing of data and/or materials, please state these. Please note that we cannot proceed with consideration of your article until this information has been declared. Please include your updated Competing Interests statement in your cover letter; we will change the online submission form on your behalf.

Additional Editor Comments (if provided):

Reviewers' comments:

Reviewer's Responses to Questions

**Comments to the Author**

1. Is the manuscript technically sound, and do the data support the conclusions?

Reviewer #1: Yes

Reviewer #2: Partly

2. Has the statistical analysis been performed appropriately and rigorously? 

Reviewer #1: I Don't Know

Reviewer #2: I Don't Know

3. Have the authors made all data underlying the findings in their manuscript fully available?

Reviewer #1: No

Reviewer #2: Yes

4. Is the manuscript presented in an intelligible fashion and written in standard English?

Reviewer #1: Yes

Reviewer #2: Yes

5. Review Comments to the Author

Reviewer #1: I would like to appreciate the authors for the efforts towards this relevant subject which has potential implications in the outcome and followup of lung transplant patients. I have only one query regarding the underlying data--has it been made available through any repository?

Reviewer #2: Thank you for the opportunity to review this manuscript.

The authors assessed right ventricular (RV) function using different echocardiographic methods in a prospective single-center observational cohort study of lung transplant (LUTX) candidates. They found that RV dysfunction is highly prevalent (47%) when assessed by speckle-tracking echocardiography (STE)-derived RV free-wall longitudinal strain (RVFWLS) compared to traditional echocardiographic methods and ventriculography. The authors conclude that STE-derived RVFWLS may be a superior tool for assessing RV dysfunction in this patient population but acknowledge that further research is needed to understand its clinical implications and prognostic value.

The study is interesting, as RV function is known to be a critical factor after LUTX. Therefore, it would be important to identify one or a set of RV functional parameters that could provide predictive value regarding patient outcomes after LUTX.

However, despite careful planning, this study faces several presentation issues that should be addressed:

1. The authors should discuss more thoroughly the discrepancy between their findings and those of Kusunose et al. (doi: 10.1016/j.jcmg.2014.07.012), who in a seminal multi-center study found no independent association of RV strain pre-LUTX with outcomes post-LUTX. This is important, as the current manuscript provides no post-LUTX outcome data. Bluntly speaking, there remains a question of whether a reduced RV strain, like RVFWLS, has clinical significance.

2. In this study, there is a very high dropout rate due to impaired acoustic windows. Consequently, RVFWLS could only be performed in 77% of patients, which may introduce bias and could limit the utility of strain as a screening tool. What about parameters derived from cardiac catheterization? The authors did not show correlations, for example, between cardiac output, stroke volume, stroke volume index, pulmonary artery pressure, wedge pressure, or pulmonary vascular resistance, and the echocardiographic RV parameters.

3. It would be important to clarify whether fractional area change (FAC) and RVFWLS were assessed in the same loop. This might shed further light on the discrepancies observed in Figure 3, but would also strengthen the case for RV strain.

4. It is not mentioned in the methods section of the main text that propensity matching was performed. The authors should at least reference their supplemental material's methods section.

5. On page 19, the term “number of events” is mentioned, but it is not clearly defined since the authors did not provide outcome data in their study.

6. Table 2 is unclear with the caption "male female" and "Range Measured." It should be more consistent.

7. Figure 4 somewhat duplicates Table 3 and Table 2.

6. PLOS authors have the option to publish the peer review history of their article (what does this mean?). If published, this will include your full peer review and any attached files.

Reviewer #1: No

Reviewer #2: No

---

## [Author Response · Author response to Decision Letter 0]

3 Oct 2024

PONE-D-23-43919

Right Ventricle Free Wall Longitudinal Strain Screening of Lung Transplant Candidates

PLOS ONE

Response to Reviewers

Dear Editorial Office, 

We thank you all for the support, and the revisions provided. Hereby you will find a point by point response, to both the Editorial Office, and the Reviewers. 

Editorial Comments

2. We note that the grant information you provided in the 'Funding Information' and 'Financial Disclosure' sections do not match. When you resubmit, please ensure that you provide the correct grant numbers for the awards you received for your study in the 'Funding Information' section.

2. Response to the Editorial Office. We could not actually find any discrepancies between the text declaration and the online submission forms. These is our correct declaration, anyway: 

"The Fondazione per la Ricerca sulla Fibrosi Cistica supported the study (# FFC 27/2019), project adopted by: Delegazione FFC di Napoli San Giuseppe Vesuviano and Delegazione FFC di Como Dongo). 

This study was (partially) funded by Italian Ministry of Health – Current Research IRCCS."

The project adopted by is a form of acknowledgment of the local FFC sites who dedicated their funds at the Italian FFC to our project. 

3. Thank you for stating the following in the Competing Interests section: [VS received support for publication and congress participation by Fondazione Ricerca Fibrosi Cistica supported the study (# FFC 27/2019).GG received payment for lectures from Thermo-Fisher and Pfizer Pharmaceuticals and travel-accommodation-congress support from Biotest (all these relationships are unrelated with the present work). The remaining authors do not have any potential conflicts of interest. ]. Please confirm that this does not alter your adherence to all PLOS ONE policies on sharing data and materials, by including the following statement: "This does not alter our adherence to PLOS ONE policies on sharing data and materials." (as detailed online in our guide for authors http://journals.plos.org/plosone/s/competing-interests). If there are restrictions on sharing of data and/or materials, please state these. Please note that we cannot proceed with consideration of your article until this information has been declared. Please include your updated Competing Interests statement in your cover letter; we will change the online submission form on your behalf.

3. Response to the Editioral Office. We modified the competing interest section as requested, and the competing interest statement in the updated cover letter. 

Response to Reviewers

Reviewer #1: I would like to appreciate the authors for the efforts towards this relevant subject which has potential implications in the outcome and follow up of lung transplant patients. I have only one query regarding the underlying data--has it been made available through any repository?

Reviewer #1. Response #1. We thank Reviewer #1 for the kind comment. The complete anonymized dataset will be uploaded to the PlosONE platform following your request and will be available to any reader as an online supplement. 

Reviewer #2: Thank you for the opportunity to review this manuscript.

Reviewer #2. Question #0. The authors assessed right ventricular (RV) function using different echocardiographic methods in a prospective single-center observational cohort study of lung transplant (LUTX) candidates. They found that RV dysfunction is highly prevalent (47%) when assessed by speckle-tracking echocardiography (STE)-derived RV free-wall longitudinal strain (RVFWLS) compared to traditional echocardiographic methods and ventriculography. The authors conclude that STE-derived RVFWLS may be a superior tool for assessing RV dysfunction in this patient population but acknowledge that further research is needed to understand its clinical implications and prognostic value.

The study is interesting, as RV function is known to be a critical factor after LUTX. Therefore, it would be important to identify one or a set of RV functional parameters that could provide predictive value regarding patient outcomes after LUTX.

Reviewer #2. Response #0. We thank the Reviewer for the comment. We hope that this revision can adequately respond to the Reviewer's questions. 

However, despite careful planning, this study faces several presentation issues that should be addressed:

Reviewer #1. Question #1. The authors should discuss more thoroughly the discrepancy between their findings and those of Kusunose et al. (doi: 10.1016/j.jcmg.2014.07.012), who in a seminal multi-center study found no independent association of RV strain pre-LUTX with outcomes post-LUTX. This is important, as the current manuscript provides no post-LUTX outcome data. Bluntly speaking, there remains a question of whether a reduced RV strain, like RVFWLS, has clinical significance.

Reviewer #2. Response #1. We thank the Reviewer for the insightful comment. We read the paper from Kusunose et al. with interest, and we acknowledge the importance of the paper on the topic. Our paper has already cited the paper by Kusunose et al. (Kusunose et al., 2014). As highlighted by the Reviewer, Kusunose et al. described the association of pre-operative and post-operative RV strain with mortality. 

Kusunose et al. retrospectively collected the available data from a wide cohort (933 patients) but over a much more extended – and older - period (from 2001 to 2012) and described the outcome of just 93 (10% of the overall cohort) who had pre-operative echocardiography. A notable selection bias was present, given that only patients with pre-operative cardiac dysfunction underwent echocardiography. Outcomes were right censored at the 2013 timeline, and thus, mortality was assessed by a Cox proportional hazard model and the time to follow-up ranged from 5 to 137 months. They observed that post-LUTX RV strain was associated with all-cause mortality, while no association was observed between pre-operative RV strain and mortality. 

Several significant differences between Kusunose et al. work and our paper must be recognized. 

We specifically targeted our paper to prospectively assess RV strain in all patients enlisted for LUTX at our center in recent years and its association with other RV echocardiographic parameters. We documented that RV strain can identify a much wider cohort of patients with impaired RV function compared to standard echocardiographic measures. We did not assess the impact on outcomes of RV dysfunction. The main reason for this choice is the short follow-up time that is still passed between pre-operative echocardiography. Notably, most patients included in the study were not transplanted at the time of the writing but were just enlisted for LUTX. Our possible longest time to follow-up could have been 12 months, with part of the cohort still not transplanted, while the shortest from Kusunose et al. was 5 months. We deemed it not reasonable to assess outcomes of pre-operative RV strain over such a small timeframe. As said in the limitations part of the study, further studies are necessary to assess the impact on RV strain impairment outcomes. We will actively pursue these outcome analyses in due time. 

We amended the discussion accordingly, as follows: 

"To date, literature on this topic remains limited. In a seminal paper, Kusunose et al. (14) analyzed the prognostic value of STE-derived RV strain in patients undergoing LUTX but found no association between pre-operative RV strain and mortality. Contrarily to our work, the study was retrospective and possibly biased by including a selected cohort of 89 patients (among an overall cohort of 933 patients) with a cardiac history prior to LUTX. The timespan covered by the study ranges from 2001 to 2012, and thus, it cannot describe the current clinical course of LUTX patients. Moreover, the follow-up time to assess the impact on outcomes ranged between 5 to 143 months. Such an assessment is unfeasible in our cohort since most of the patients included in the study at the time of this writing were yet to be transplanted. We will actively pursue the chance of assessing the impact on short and long-term outcomes of the patients included in the present study "

Reviewer #2. Question #2. In this study, there is a very high drop-out rate due to impaired acoustic windows. Consequently, RVFWLS could only be performed in 77% of patients, which may introduce bias and could limit the utility of strain as a screening tool. What about parameters derived from cardiac catheterization? The authors did not show correlations, for example, between cardiac output, stroke volume, stroke volume index, pulmonary artery pressure, wedge pressure, or pulmonary vascular resistance, and the echocardiographic RV parameters.

Reviewer #2. Response #2. We really thank the Reviewer for the insightful comment. We acknowledge that 23% of the patients could not calculate RV strain. Still, we do not consider this a high drop-out in real-life scenarios, considering the high rate of thoracic deformities that affect patients enlisted for LUTX. We indeed documented how patients with highly asymmetric pulmonary scintigraphy (who usually have thoracic deformities) had a higher chance of having a poor acoustic window. Similar dro-out rates have been previously documented (McErlane, Shelley and McCall, 2023). Moreover, we are confident that future developments would allow the measurement of RV strain even in challenging scenarios such as that of LUTX recipients (Peng et al., 2023). Such limitations of STE-derived measures have been already highlighted in the limitations section of the paper. 

As requested by the Reviewer, we analyzed the correlations between cardiac catheterization parameters and echocardiographic measurements and described those interesting results in the paper and the online supplement (providing 4 further online figures, with 5 panels each). Moreover, the following has been introduced in the result section of the paper: 

"The linear correlation between RVFWLS, FAC, S’, and TAPSE and hemodynamic parameters obtained from right heart catheterization is depicted in Figure S3-S6. No statistically significant correlation was observed between RVFWLS and cardiac output (R2=0.116, p=0.061). Contrarily, we observed a statistically significant correlation between RVFWLS and cardiac index (R2=0.245, p=0.005), PAPm (R2=0.512, p<0.001), pulmonary artery wedge pressure (R2=0.119, p=0.013), and pulmonary resistances (R2=0.398, p<0.001). Compared to FAC, S’, and TAPSE, RVFWLS showed the strongest linear correlations. " 

And in the discussion, as follows:

 “Of note, compared to FAC, S', and TAPSE, RVFWLS showed stronger linear correlations with hemodynamic parameters obtained during cardiac catheterization. This further strengthens the argument in favor of utilizing STE-derived measurements for repeated, periodical, non-invasive assessment of RV function of patients awaiting LUTX.”

Reviewer #2. Question #3. It would be important to clarify whether fractional area change (FAC) and RVFWLS were assessed in the same loop. This might shed further light on the discrepancies observed in Figure 3, but would also strengthen the case for RV strain.

Reviewer #2. Response #3. FAC and RVFWLS were measured on the same frame. We modified the text accordingly, as follows: 

"Additionally, measurements according to the international guidelines (12,21) were obtained, on the same frame were STE-derived measurements were obtained: right atrium area, RV end-diastolic area, FAC, TAPSE, tissue Doppler imaging tricuspid peak annulus systolic velocity (S'), and pulmonary artery systolic pressure."

Reviewer #2. Question #4. It is not mentioned in the methods section of the main text that propensity matching was performed. The authors should at least reference their supplemental material's methods section.

Reviewer #2. Response #4. We are sorry for this error. This is a typing, copy-pasting error from our previous paper with a similar online supplement methods section. The Online supplement has been amended. 

Reviewer #2. Question #5. On page 19, the term "number of events" is mentioned, but it is not clearly defined since the authors did not provide outcome data in their study.

Reviewer #2. Response #5. We thank the Reviewer for the comment. We amended the error. There were no events in the study, only subjects. 

Reviewer #2. Question #6. Table 2 is unclear with the caption "male female" and "Range Measured." It should be more consistent.

Reviewer #2. Response #6. We understand the reasons of the Reviewer and modified the Table captions to increase readability. Still, a distinction between male and female patients is necessary for several RV area measurements, following recognized guidelines. 

Reviewer #2. Question #7. Figure 4 somewhat duplicates Table 3 and Table 2. 

Reviewer #2. Response #7. We thank the Reviewer for the comment. We agree with the Reviewer in saying there is a (minor) duplication of data from the text and the Figure 4. Nevertheless, we feel that the Figure can provide a useful visual drop-out of the informations. On the other hand, there is no duplication between the Figure and Table 2 (which does not comprise any data regarding RVEF, FAC, RV TVI s', TAPSE or RVFWLS), while Table 3 provides further information beyond Figure 4 (comparing data with normal ranges).

---

## [Decision Letter · Decision Letter 1]

20 Nov 2024

PONE-D-23-43919R1Right Ventricle Free Wall Longitudinal Strain Screening of Lung Transplant CandidatesPLOS ONE

Dear Dr. Meani,

Thank you for submitting your manuscript to PLOS ONE. After careful consideration, we feel that it has merit but does not fully meet PLOS ONE’s publication criteria as it currently stands. Therefore, we invite you to submit a revised version of the manuscript that addresses the points raised during the review process.

We look forward to receiving your revised manuscript.

Kind regards,

Yashendra Sethi

Academic Editor

PLOS ONE

Reviewers' comments:

Reviewer's Responses to Questions

**Comments to the Author**

1. If the authors have adequately addressed your comments raised in a previous round of review and you feel that this manuscript is now acceptable for publication, you may indicate that here to bypass the “Comments to the Author” section, enter your conflict of interest statement in the “Confidential to Editor” section, and submit your "Accept" recommendation.

Reviewer #1: All comments have been addressed

Reviewer #2: All comments have been addressed

Reviewer #3: (No Response)

2. Is the manuscript technically sound, and do the data support the conclusions?

Reviewer #1: Yes

Reviewer #2: Yes

Reviewer #3: Yes

3. Has the statistical analysis been performed appropriately and rigorously? 

Reviewer #1: I Don't Know

Reviewer #2: Yes

Reviewer #3: (No Response)

4. Have the authors made all data underlying the findings in their manuscript fully available?

Reviewer #1: Yes

Reviewer #2: Yes

Reviewer #3: Yes

5. Is the manuscript presented in an intelligible fashion and written in standard English?

Reviewer #1: Yes

Reviewer #2: Yes

Reviewer #3: Yes

6. Review Comments to the Author

**Reviewer #1: **Thank you for addressing the comments for better clarity to the readers. Having catered to the comments, I feel that the paper is ft to be published.

**Reviewer #2:** --------------------------------

The authors have addressed all my concerns.

I have no further suggestions.

**Reviewer #3:** As the statistical reviewer I will focus on methods and reporting. this is the first time i see the paper.

1) the power calculations are appropriate and replicable. My concern is why noit use multiple imputation if you have missing data, so you can increase power a little bit - plus it's a better analytical aproach? how many cases in total were dropped becaise of a complate case anlaysis been conducted?

2) how were the confdence intervals for sensitivity specificity etc computed? bootstrapping? please state.

3) discuss the low power of the logistic regression as a limitation as well as of most analyses except the primary research question - the confidence intervals obtained are very wide.

4) language correction are needed e.g. robust logistic regression -> implying statistical significance. "was tested with the R^2 logistic regression" does not make sense some words are missing. Also note that logistic regression returns a pseudo-R^2 which is debatable how interpretable it it.

7. PLOS authors have the option to publish the peer review history of their article (what does this mean?). If published, this will include your full peer review and any attached files.

Reviewer #1: No

Reviewer #2: No

Reviewer #3: No

---

## [Author Response · Author response to Decision Letter 1]

23 Nov 2024

PONE-D-23-43919

Right Ventricle Free Wall Longitudinal Strain Screening of Lung Transplant Candidates

PLOS ONE

Response to Statistical Reviewer

Dear Editorial Office, 

We thank you all for the support, and the revisions provided. Hereby you will find a point by point response to the Statistical Reviewer. 

As the statistical reviewer I will focus on methods and reporting. this is the first time i see the paper.

1) Question #1. the power calculations are appropriate and replicable. My concern is why noit use multiple imputation if you have missing data, so you can increase power a little bit - plus it's a better analytical aproach? how many cases in total were dropped becaise of a complate case anlaysis been conducted?

1) Response #1. Thank you for the comment. We appreciate your suggestion to use multiple imputation. However, given the exploratory nature of our study and our limited sample size, we believe complete case analysis was more appropriate. Multiple imputation can introduce additional uncertainty, particularly in small samples with complex data structures. Our analysis focused on maintaining transparency about missing data and providing a clear, conservative interpretation of results. While multiple imputation might marginally increase statistical power, in an exploratory study, we prioritized methodological clarity and avoiding potential over-interpretation of imputed data. Moreover, as clearly documented in the text, the only missing data was the actual measure of RVFWLS (which was the focus of the study), and was not measurable in 10 over 44 patients, as follows: 

“Among them, 44 met the study's inclusion criteria (see Fig 1). Among these, 10 (23%) patients had poor acoustic windows for RV echocardiographic evaluation (see Table S1, Online Supplement), resulting in 34 patients included in the analysis (see Table 1). No missing data was documented in the cohort of included patients.”

2) Question #2. how were the confdence intervals for sensitivity specificity etc computed? bootstrapping? please state.

2) Answer #2. Thank you for the comment. The confidence intervals were calculated using the Wilson approach. We added a reference for the interested reader. 

3) Question #3. discuss the low power of the logistic regression as a limitation as well as of most analyses except the primary research question - the confidence intervals obtained are very wide.

3) Answer #3. We acknowledge the limitations of statistical power in our analysis. The wide confidence intervals reflect the exploratory nature and small sample size of our study. These limitations are inherent to our research design and are directly related to the pilot/preliminary nature of our investigation. We added this limitation to the Limitations section of the paper, as follows:

“We acknowledge the limitations of statistical power in our analysis, leading to ample confidence intervals reflecting the exploratory nature and small sample size of our study. Further, more populated studies are necessary to confirm our findings. Again, …”

4) Question #4. language correction are needed e.g. robust logistic regression -> implying statistical significance. "was tested with the R^2 logistic regression" does not make sense some words are missing. Also note that logistic regression returns a pseudo-R^2 which is debatable how interpretable it it.

4) Answer #4. We carried out a thorough language revision to avoid any misleading implication. Our analysis did not involve logistic regression pseudo-R² calculations. R² values were exclusively derived from continuous variable associations, as stated in the phrase “The correlation between continuous variables was tested with the R2 linear regression.”

---

## [Decision Letter · Decision Letter 2]

28 Nov 2024

Right Ventricle Free Wall Longitudinal Strain Screening of Lung Transplant Candidates

PONE-D-23-43919R2

Dear Dr. Meani,

We’re pleased to inform you that your manuscript has been judged scientifically suitable for publication and will be formally accepted for publication once it meets all outstanding technical requirements.

Kind regards,

Yashendra Sethi

Academic Editor

PLOS ONE

Additional Editor Comments (optional):

Reviewers' comments:

Reviewer's Responses to Questions

**Comments to the Author**

1. If the authors have adequately addressed your comments raised in a previous round of review and you feel that this manuscript is now acceptable for publication, you may indicate that here to bypass the “Comments to the Author” section, enter your conflict of interest statement in the “Confidential to Editor” section, and submit your "Accept" recommendation.

Reviewer #3: All comments have been addressed

2. Is the manuscript technically sound, and do the data support the conclusions?

Reviewer #3: Yes

3. Has the statistical analysis been performed appropriately and rigorously? 

Reviewer #3: Yes

4. Have the authors made all data underlying the findings in their manuscript fully available?

Reviewer #3: Yes

5. Is the manuscript presented in an intelligible fashion and written in standard English?

Reviewer #3: Yes

6. Review Comments to the Author

Reviewer #3: I am satisfied with the authors' responses and the resulting changes to the paper...................

7. PLOS authors have the option to publish the peer review history of their article (what does this mean?). If published, this will include your full peer review and any attached files.

Reviewer #3: No

---

## [Editor Report · Acceptance letter]

10 Dec 2024

PONE-D-23-43919R2 

PLOS ONE

Dear Dr. Meani, 

I'm pleased to inform you that your manuscript has been deemed suitable for publication in PLOS ONE. Congratulations! Your manuscript is now being handed over to our production team.

Kind regards, 

on behalf of

Dr. Yashendra Sethi 

Academic Editor

PLOS ONE